# Recipes for the digital quantum simulation of lattice spin systems

## Guido Burkard

Department of Physics, University of Konstanz, D-78457 Konstanz, Germany

## Abstract

We describe methods to construct digital quantum simulation algorithms for quantum spin systems on a regular lattice with local interactions. In addition to tools such as the Trotter-Suzuki expansion and graph coloring, we also discuss the efficiency gained by parallel execution of an extensive number of commuting terms. We provide resource estimates and quantum circuit elements for the most important cases and classes of spin systems. As resource estimates we indicate the total number of gates $N$ and simulation time $T$, expressed in terms of the number $n$ of spin 1/2 lattice sites (qubits), target accuracy $\epsilon$, and simulated time $t$. We provide circuit constructions that realize the simulation time $T^{(1)} \propto nt^2/\epsilon$ and $T^{(2q)} \propto t^{1+\eta}n^{\eta}/\epsilon^{\eta}$ for arbitrarily small $\eta = 1/2q$ for the first-order and higher-order Trotter expansions. We also discuss the potential impact of scaled gates, which have not yet been fully explored.

| Received | 2025-01-21 |
| Accepted | 2025-02-25 |
| Published | 2025-03-10 |

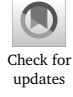

## 1 Introduction

The idea that the simulation of quantum systems, while exponentially hard on a classical computer, can be done efficiently on a quantum computer dates back three decades, to Richard

Feynman [1]. A specific task of a digital quantum simulator consists of *Hamiltonian simulation*, i.e. the propagation of an initial state $\psi(0)$ to the state $\psi(t)$ at a later time $t > 0$ according to the dynamics generated by a Hamiltonian $H(\tau)$ via the time-dependent Schrödinger equation, $i\hbar\partial_\tau\psi(\tau) = H(\tau)\psi(\tau)$, for $0 \leq \tau \leq t$. Quantum simulation has been studied for a variety of quantum systems on several simulator platforms [2]. Systems that can be simulated include quantum field theories [3], quantum chemistry [4], and fermionic lattice models [5,6]. Experimental demonstrations of quantum simulation have been realized using superconducting circuits [7,8], ion traps [9], and semiconductor spin qubits [10].

The resources required to simulate the time evolution of a discrete quantum system on a quantum computer can be quantified in terms of the system size (measured in the number of qubits $n$ required to store its quantum state), as well as the duration $t$ and desired accuracy $\epsilon$ of the simulation. For a lattice of spins 1/2, $n$ directly represents the number of lattice sites. The measure we use to quantify the simulation complexity is the duration $T = T(n, t, \epsilon)$ of the simulation, not to be confused with the simulated time $t$.

One of the early results of quantum simulation was the insight that quantum systems with local interactions can be simulated efficiently on a quantum computer using the Trotter decomposition [11]. The number $m$ of elementary, discrete time increments needed to simulate the quantum evolution of a system during time $t$ within accuracy $\epsilon$ turns out to be proportional to $t^2/\epsilon$. The elementary time increments consist of the simulation of a local interaction for a small time step $\Delta t = t/m$. Since $m \propto t^2$ the time steps scale as $\Delta t \propto 1/t$. If time increments can be implemented on a quantum computer with a native gate with a gate time proportional to the simulated time ('scaled gate'), $t_g \propto \Delta t \propto 1/t$, then the overall simulation time $T$ scales as $T = mt_g \propto t$, i.e., the simulation time $T$ is proportional to the simulated time $t$. In many cases, scaled gates may not be available, but it turns out that the method of higher-order Trotterization can approach computation times $\propto t^{1+\eta}/\epsilon^\eta$ where $\eta$ can be made arbitrarily small [12,13]. The concept of scaled gates is also related to the analog blocks in digital-analog quantum simulations [8]. Similar estimates can be made for the general class of sparse Hamiltonians [14]. It is also known that there is no sub-linear scaling of $T$ with the simulated time $t$, a restriction known as "no fast-forwarding theorem" [14].

Another question relates to the scaling of the simulation time $T$ with the size of the simulated system, e.g., measured in the number of qubits $n$ required to store its quantum state. Raeisi *et al.* [12] find that for $k$-local Hamiltonians $H = \sum_j H_j$ where each $H_j$ acts on at most $k$ qubits, $T$ asymptotically scales as $n^{(2+\eta)k-1}t^{1+\eta}/\epsilon^\eta$, again with arbitrarily small $\eta$. If the number of qubits with which each qubit can interact is bounded by a constant, the model is called physically $k$-local, and $T$ is found to scale as $n^{1+\eta}t^{1+\eta}/\epsilon^\eta$, i.e., nearly linear in both the simulated time $t$ and the system size $n$ [12].

A useful observation is that for interacting sites described by a regular graph, and in particular for lattices (such as those shown in Fig. 1), graph coloring according to Vizing's theorem allows for the decomposition of the Hamiltonian into a number $K$ of commuting parts which does not grow with the system size. This allows for efficient algorithms with simulation times asymptotically scaling as $t$, and independent of $n$, up to poly-logarithmic corrections in $nt/\epsilon$ [15]. In this paper, we study the Hamiltonian simulation of spin models on a lattice that constitutes a digital quantum simulation of physically 2-local Hamiltonians. We provide explicit algorithms with the same asymptotic scaling.

The microscopic understanding of magnetic phenomena, starting from ferromagnetism typically requires a quantum model [16]. Magnetic behavior can be modeled by spin models where spins are typically located on a lattice, see Fig. 1. The Heisenberg exchange interaction between the spins is short-range and it is often an excellent approximation to assume that only nearest neighbor spins are coupled. Hamiltonian quantum simulation of such spin models can provide useful insight into the time-dependent phenomena of magnetic systems that is often

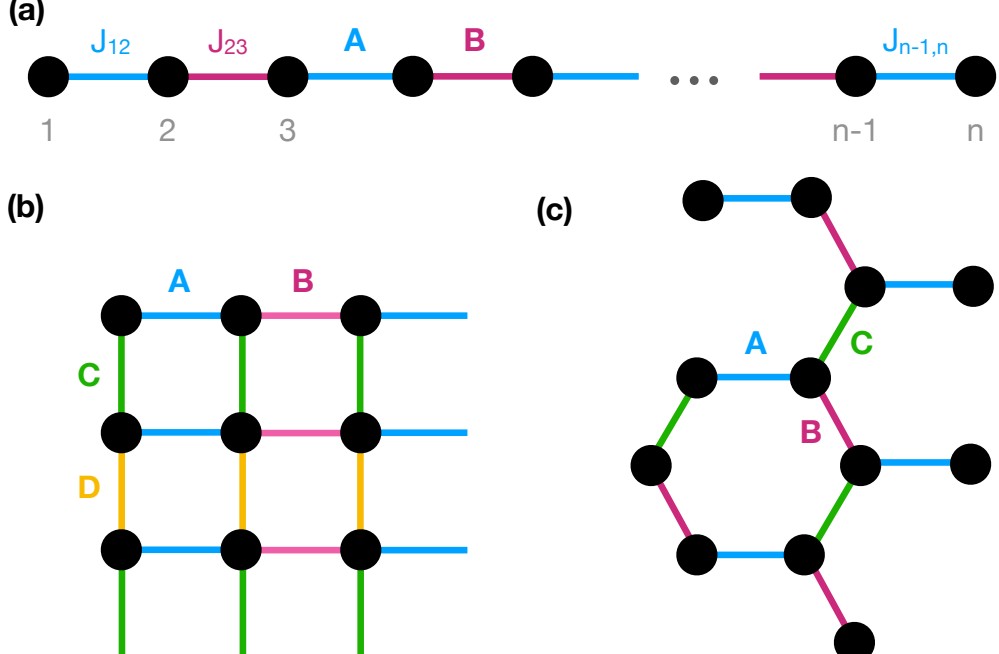

Figure 1: **Lattice spin system** with local interactions, in one (a) and two (b,c) spatial dimensions. A spin system can be described as a graph with vertices (edges) representing sites $i$ (non-zero couplings $J_{ij} \neq 0$). In a regular lattice, the coordination number and the chromatic index $K$ are equal. The coordination number counts the number of nearest neighbors, while the chromatic index of a graph is the number of colors needed to color all edges without two same-colored edges meeting at any vertex. (a) One-dimensional lattice, $K = 2$, (b) square lattice, $K = 4$, (b) hexagonal lattice, $K = 3$.

hard to compute classically, even in cases where the ground state of the system is relatively easy to obtain. Quantum circuits for the digital quantum simulation of disordered one-dimensional Heisenberg chains have been developed and their scaling in $n$ and $t$ analyzed by Childs *et al.* [13], where typically $t \propto n$ was chosen to simulate self-thermalization where information needs to propagate through the entire system.

It is known that the Heisenberg interaction and other spin-spin interactions–when combined with local coupling of individual spins to an (effective) magnetic field –generate a universal set of quantum gates for spin-1/2 qubits [17]. The Heisenberg interaction alone can generate universal quantum computing on three-spin-1/2 exchange-only or decoherence-free subspace qubits [18]. Here, we will be concerned with the opposite simulation direction, where a universal quantum computer simulates a spin system.

## 2 Spin models

To begin, we define our general spin model, describing a finite number $n$ of spins $S_i = (S_i^x, S_i^y, S_i^z)$ where $i = 1, 2, \ldots, n$ whose dynamics are described by the Hamiltonian

$$H = \sum_{i<j} H_{ij} = \sum_{i<j} \sum_{\alpha\beta} J_{ij}^{\alpha\beta} S_i^\alpha S_j^\beta + \sum_{i\alpha} h_i^\alpha S_i^\alpha, \qquad (1)$$

where the spin operators fulfill the angular momentum commutation rules, $\left[S_i^\alpha, S_j^\beta\right] = i\delta_{ij}\sum_\gamma \epsilon_{\alpha\beta\gamma}S_i^\gamma$, and where the interactions $J_{ij}^{\alpha\beta}$ and external fields $h_i^\alpha$ can be time dependent. The length of the spin is arbitrary at this point but we will later choose $S = 1/2$ where each spin can be represented by one qubit. We choose units in which $\hbar = 1$ throughout this paper. The site indices $i$ and $j$ run from 1 to the number of sites $n$, and the Cartesian coordinate indices $\alpha$, $\beta$, and $\gamma$ take the values $x$, $y$, and $z$ for a three-dimensional spin. The term $H_{ij}$ is defined such that it contains only spin operators $\mathbf{S}_i$ and $\mathbf{S}_j$; such that we can, e.g., include the $h_i^\alpha$ terms in $H_{i,i+1}$ for $i < n$ and in $H_{n-1,n}$ for $i = n$. Note that $\left[H_{ij}, H_{i'j'}\right] = 0$ for disjoint pairs $\{i,j\} \cap \{i',j'\} = \emptyset$. The connectivity graph defined by all nonzero $J_{ij}$ tensors is completely general at this point, but will be restricted below to regular physical lattices that are constrained by locality and spatial dimension. The isotropic Heisenberg model represents an important special case where $J_{ij}^{\alpha\beta} = \delta_{\alpha\beta}J_{ij}$ and thus

$$H = \sum_{i<j} J_{ij}\mathbf{S}_i \cdot \mathbf{S}_j + \sum_i \mathbf{h}_i \cdot \mathbf{S}_i. \tag{2}$$

The Hamiltonian generates the time evolution of the spin system in the form of a time-ordered exponential

$$\psi(t) = \mathrm{T}\exp\left(-i\int_0^t H(\tau)d\tau\right)\psi(0) = U(t)\psi(0), \tag{3}$$

which can be approximated as a product of a finite number $m$ of simple operator exponentials, $U(t) \approx \prod_{p=0}^{m-1} \exp\{-i(t/m)H(pt/m)\}$ with the earliest times appearing on the right of the product. If the Hamiltonian is time-independent, one has $U(t) = \exp(-itH)$, but this will not be assumed here. In general, these exponentials cannot be decomposed into factors $\exp(-itH_{ij})$ operating on spin pairs because various pairs of terms in $H$ do not commute. To approximate $U(t)$, we divide $H$ into a sum of $K$ non-commuting parts $H_k$ each of which consists only of commuting terms,

$$H = \sum_{k=1}^K H_k, \qquad H_k = \sum_{(i,j)\in P_k} H_{ij}, \tag{4}$$

with $[H_k, H_l] \neq 0$ for $k \neq l$. Here, the sets $P_k$ are chosen such that they do not contain any common spins, and thus $\left[H_{ij}, H_{nm}\right] = 0$ for $(i,j),(n,m) \in P_k$, and thus

$$\exp(-i\tau H_k) = \prod_{(i,j)\in P_k} \exp(-i\tau H_{ij}) = \prod_{(i,j)\in P_k} U_{ij}. \tag{5}$$

The problem of finding a minimal number $K$ of sets $P_k$ with this property is equivalent to the edge coloring problem of the graph $G$ consisting of $n$ vertices (one for each spin) and an edge for each non-zero $J_{ij}$. In graph theory, the minimal $K$ is referred to as the chromatic index (Fig. 1). Vizing's theorem [12] states that for a graph $G$ of degree $\deg(G)$, one has $\deg(G) \leq K \leq \deg(G) + 1$. For bipartite lattices, $K = \deg(G)$. Here, the degree $\deg(G)$ of $G$ is defined as the maximum number of edges containing the same vertex, i.e., the maximum number of spins coupled to one and the same spin. In the case of all-to-all coupling where all $n(n-1)/2$ possible couplings between spins are assumed to be nonzero, the graph describing the spin model is the complete graph with degree $n-1$ and $K = n-1$ if $K$ is even and $K = n$ if $n$ is odd. In both cases, we find $K = O(n)$.

In the following, we will study spin models on a regular lattice with local interactions, with each spin (in the bulk) being coupled to at most its $\deg(G) = z$ adjacent spins where the coordination number $z$ of the lattice is independent of $n$. Examples of lattice spin models in one and two dimensions are shown in Fig. 1. In this case $K = z$ and we will use the notation $K$ for both the coordination number and chromatic index.

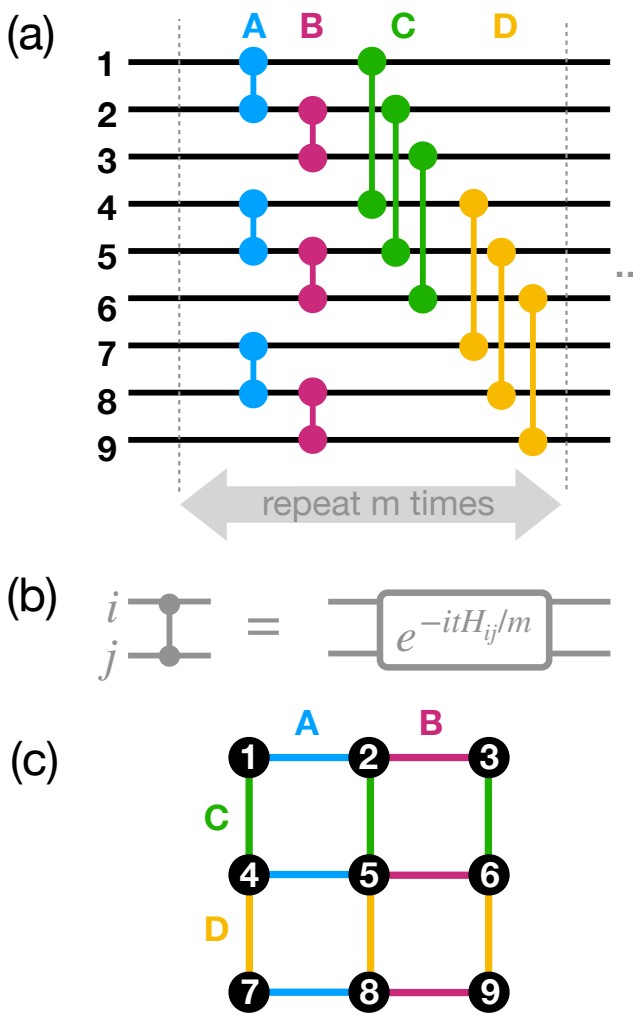

Figure 2: **Quantum circuit for the quantum simulation of a spin system.** (a) Quantum circuit for the Hamiltonian simulation of the lattice spin system shown in (c) with $K = 4$ within the first-order Trotterization. Colors and labels A, B, C, and D represent corresponding edges in the lattice. The depicted sequence is repeated $m$ times where $m \sim K^2 t^2 n J^2$. (b) Elementary time increment simulating the interaction between spins $i$ and $j$ during the time $t/m$.

## 3 First-order Trotter-Suzuki expansion

The non-commutativity of the $H_k$ can be dealt with using the Trotter-Suzuki expansion. The first-order Trotter-Suzuki formula [19,20] allows for a digital quantum simulation using alternating simulations of the non-commuting parts $H_k$ of $H = \sum_k H_k$,

$$e^{-it \sum_k H_k} = \lim_{m \to \infty} \left( \prod_{k=1}^{K} e^{-itH_k/m} \right)^m = \lim_{m \to \infty} S^{(1)}(t,m)^m . \tag{6}$$

A practical quantum simulation will use only a finite number $m$ of interactions and will thus incur an error

$$\Delta_K^{(1)}(t,m) = \left\| e^{-it \sum_k H_k} - \left( \prod_{k=1}^{K} e^{-itH_k/m} \right)^m \right\| = \frac{t^2}{2m} \left\| \sum_{k<l} [H_k, H_l] \right\| + O\left( \left( \frac{t}{m} \right)^3 \right), \tag{7}$$

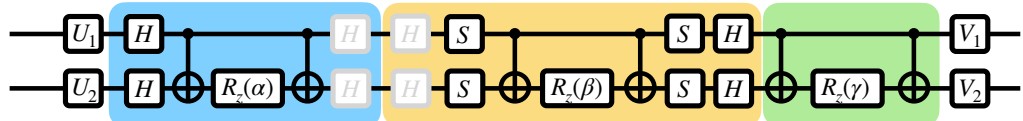

Figure 3: **Quantum circuit for a general two-spin interaction** $U = V_1 \otimes V_2 \exp(-i\alpha S_1^x S_2^x - i\beta S_1^y S_2^y - i\gamma S_1^z S_2^z)U_1 \otimes U_2$ using controlled-NOT (CNOT) and single-qubit gates. The blue, yellow, and green shaded sections implement the commuting operations $\exp(-i\alpha S_1^x S_2^x)$, $\exp(-i\beta S_1^y S_2^y)$, and $\exp(-i\gamma S_1^z S_2^z)$, respectively. The greyed-out Hadamard gates can be omitted as they cancel each other.

where the higher-order contributions can be more precisely written as an exponential [21]. Using the upper bound $\left\| \sum_{k<l} [H_k, H_l] \right\| \le \sum_{k<l} \| [H_k, H_l] \| \le \frac{K(K-1)}{2} \max_{k<l} \| [H_k, H_l] \|$ and evaluating the commutators for the spin model Eqs. (1) and (4), $[H_k, H_l] = i \sum_i J_{ii_k} J_{ii_l} \sum_{\alpha\beta\gamma} \epsilon_{\alpha\beta\gamma} S_i^\alpha S_{i_k}^\beta S_{i_l}^\gamma$, where $i_k$ is the unique site such that $(i, i_k) \in P_k$, we find $\| [H_k, H_l] \| \le \frac{6}{8} n J^2$, and

$$\Delta_K^{(1)}(t, m) \le \frac{3}{4} \frac{t^2}{2m} \frac{K(K-1)}{2} n J^2, \tag{8}$$

where $J$ denotes an upper bound on the interaction strengths, $J_{ij} \le J$. In order to suppress the error below $\epsilon$, such that $\Delta_K^{(1)}(t, m) \le \epsilon$, it is thus sufficient to choose a sufficiently fine discretization of time, such that

$$m \ge \frac{3}{16} K(K-1) \frac{t^2}{\epsilon} n J^2. \tag{9}$$

The required number of elementary spin-spin coupling operations $U_{ij} = \exp(-i\tau H_{ij})$ within the first-order Trotter-Suzuki expansion can then be given as

$$N^{(1)} = m \frac{nK}{2} = \frac{3}{32} K^2(K-1) \frac{t^2}{\epsilon} n^2 J^2. \tag{10}$$

A quantum circuit realizing the digital quantum simulation of a spin system is shown in Fig. 2. The circuit size (gate count) is proportional to $N^{(1)}$; as we show below, the CNOT count for general spin-spin interactions amounts to $6N^{(1)}$, which is reduced to $3N^{(1)}$ in the case of Heisenberg interactions. To quantify the circuit depth (simulation time) we observe that the elementary spin-spin coupling operations $U_{ij} = \exp(-i\tau H_{ij})$ inside each $H_k$ can be executed in parallel (Fig. 2a). Therefore, we find for the simulation time,

$$T^{(1)} = mKt_g = \frac{3}{16} K^2(K-1) \frac{t^2}{\epsilon} n J^2 t_g, \tag{11}$$

where $t_g$ is the maximum time required to execute $U_{ij} = \exp(-i\tau H_{ij})$ with the simulated time increment $\tau = t/m$. The quantum circuit for $U_{ij}$ may contain 'fixed' quantum gates that require a gate time that is independent of $\tau$, others may be 'scaled', i.e., require a gate time proportional to $\tau$ (see also the concept of digital-analog simulation [8]). In the case of a digital quantum simulation, there are some fixed gates, such as CNOT, and thus $t_g = t_\infty + st/m$ with $t_\infty > 0$, $s \ge 0$. Assuming that some of the used quantum gates (e.g., CNOT, H, etc.) have a fixed gate time, and noting that for large $m$ the contribution of scaled gates to the gate time is small and can be bounded by a constant, we set $s = 0$ and $t_\infty > 0$, and thus $t_g = t_\infty = \text{const.}$

The results for the circuit depth and simulation run time, Eqs. (10) and (11), do not achieve the best possible scaling in $n$ and $t$. In the following two sections, we discuss two possibilities to further improve the scaling. On the one hand, one can resort to higher-order Trotter-Suzuki expansions within digital quantum simulation. On the other hand, if scaled gates are available, one can proceed without the use of higher-order Trotter-Suzuki expansions.

## 4 Higher-order Trotter-Suzuki expansion

The second-order and higher-order Trotter-Suzuki formulas [22] can be written as

$$S^{(2)}(t,m) = \prod_{k=1}^{K} e^{-itH_k/2m} \prod_{k=K}^{1} e^{-itH_k/2m} \,, \tag{12}$$

$$S^{(2q)}(t,m) = \left(S_{2p-2}(p_q t,m)\right)^2 S_{2p-2}((1-4p_q)t,m)\left(S_{2p-2}(p_q t,m)\right)^2 \,, \tag{13}$$

with $p_q = (4-4^{1/(2q-1)})^{-1}$ for $q > 1$. As in the first-order case, $\lim_{m\to\infty} S^{(2q)} = e^{-it\sum_k H_k}$, for all $q \geq 1$, but the convergence becomes faster for higher orders of the Trotter-Suzuki formula. This leads to improved error bounds which can be found in Refs. [21, 23, 24],

$$\Delta_K^{(2q)}(t,m) = \left\| e^{-it\sum_k H_k} - \left(S^{(2q)}(t,m)\right)^m \right\| \tag{14}$$

$$= c_1 \frac{t^{2q+1}}{m^{2q}} \sum_{k_1,\ldots,k_{2q+1}}^{K} \left\| \left[H_{k_{2q+1}},\ldots\left[H_{k_2},H_{k_1}\right]\right]\right\| \,, \tag{15}$$

where $c_1$ is a constant that can depend on $q$ (similarly for all $c_i$ below). Evaluating the commutators for a local spin Hamiltonian, we find for the error

$$\Delta_K^{(2q)}(t,m) \leq c_2 \frac{(Kt)^{2q+1}}{m^{2q}} n \,. \tag{16}$$

Keeping the error below $\epsilon$ then requires,

$$m \geq c_3 \frac{(Kt)^{1+1/2q}}{\epsilon^{1/2q}} n^{1/2q} \,, \tag{17}$$

which leads to an interaction gate count of

$$N^{(2q)} = c_4 m \frac{nK}{2} \gtrsim c_5 \frac{n^{1+1/2q} K^{2+1/2q} t^{1+1/2q}}{\epsilon^{1/2q}} \,. \tag{18}$$

Again assuming that the interaction gates simulating $H_{ij}$ inside each $H_k$ can be executed in parallel, we find for the simulation time,

$$T^{(2q)} = c_4 m K t_g \gtrsim c_6 \frac{K^{2+1/2q} t^{1+1/2q}}{\epsilon^{1/2q}} n^{1/2q} t_g \,. \tag{19}$$

The fact that both the number of interaction gates $N^{(2q)}$ and the simulation time $T^{(2q)}$ can in principle be made to scale arbitrarily close to linearly in the simulated time $t$, has been pointed out in Ref. [14]. Also the vanishing influence of the target accuracy $\epsilon$ on the simulation time with increasing order has been recognized. The fact that the exponent of the problem size $n$ can also be suppressed is specific to constructions where the interaction gates for commuting interactions [15], as illustrated in Fig. 2.

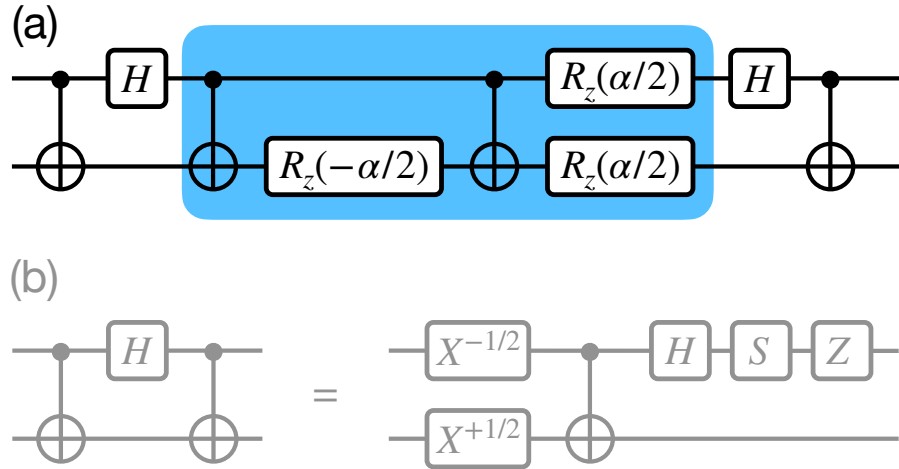

Figure 4: **Quantum circuit for the Heisenberg interaction.** (a) Implementation of $\exp(-i\alpha \mathbf{S}_1 \cdot \mathbf{S}_2) = \text{SWAP}^{\alpha/\pi}$ using controlled-NOT (CNOT) and single-qubit gates. The blue-shaded section implements a controlled-phase gate $\text{CPHASE}(\alpha)$. (b) Replacement for first three gates in (a) that eliminates one CNOT gate.

## 5 Implementation

To realize elementary interaction gates $U_{ij}$, we use the decomposition [25] $U = \exp(-i\tau H_{ij}) = V_1 \otimes V_2 \exp(-i\alpha S_1^x S_2^x - i\beta S_1^y S_2^y - i\gamma S_1^z S_2^z) U_1 \otimes U_2$, where $H_{ij} = \sum_{\alpha\beta} J_{ij}^{\alpha\beta} S_i^\alpha S_j^\beta + \sum_\alpha (h_i^\alpha S_i^\alpha + h_j^\alpha S_j^\alpha)$, and where $U_{1,2}$ and $V_{1,2}$ are one-qubit gates. This unitary can be assembled using elementary gates with the circuit shown in Fig. 3, requiring six CNOT gates. The special case of Heisenberg interactions allows for a simpler circuit with only three CNOT gates, as shown in Fig. 4.

The gate implementations comprise both fixed-length gates such as Hadamard and CNOT, and the scaled gate $R_z(\alpha)$ where $\alpha \propto \tau$ for an interval $\tau \propto 1/m$ of simulated time. With increasing number $m$ of time intervals, the fixed-length gates will dominate the execution time of such interaction gate constructions.

Depending on the quantum hardware, the interaction gates $U = \exp(-i\tau H_{ij})$ for some $H_{ij}$ may be native, i.e., directly implementable in time proportional to $\tau$, similar to the analog blocks in digital-analog simulation [8]. We call this implementation a *scaled gate*. If all required interaction gates are available as scaled gates, then $t_g = st/m$ with $s$ a constant and $t_\infty = 0$, and Eq. (11) turns into

$$T^{(1)} \leq Kst, \tag{20}$$

with $K$ and $s$ constants describing the degree of the lattice graph and the ratio between simulation time and simulated time for the slowest scaled gate. In this case, we obtain a simulation time linear in the simulated time already using the first-order Trotter expansion. The higher-order Trotter expansions do not provide any improvement in this case. The exclusive use of scaled gates renders the simulation time $T$ independent of the number $m$ of discrete time steps. Therefore, at least for time-independent problems, one can choose $m = 1$ which corresponds to a direct analog quantum simulation.

# 6  Conclusions

We have shown explicit circuit constructions that realize the resource estimates for first-order and higher-order Trotter-Suzuki product formulas. We conclude with an open question. Given a set of available native one- and two-qubit scaled gates, which other scaled gates can be efficiently constructed with this set? Do universal sets of scaled gates exist that allow for the synthesis of arbitrary scaled gates $U_{ij}$? Answering these questions may give further insight into which simulation tasks can be performed even more efficiently than shown here.

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
