# Peer review of "Recipes for the Digital Quantum Simulation of Lattice Spin Systems"

_SciPost Physics Core, doi:SciPost Phys. Core 8, 030 (2025)_

## Round 1 · Referee Report · Anonymous (Referee 1) · 2025-2-12

Strengths

The topic of digital quantum simulation is enhanced via the analysis of strategies for scaling up the simulated models via commuting terms which can be implemented in parallel, perhaps by means of native interactions which are named scaled gates here. Moreover, a novelty I find here is the time of the simulator analysis, which is interesting, especially in terms of the mentioned "scaled gates" which can reduce the implementation time of the simulator. This is also related to the analog blocks in digital-analog quantum simulations referred to in Ref. (8) in the submitted version, and perhaps this could be more explicitly mentioned, even though I think that the time analysis is rather new in this context, and possibly useful.

Weaknesses

Several typos throughout the paper should be corrected. A clearer mention of digital-analog quantum simulations as connected to the current concept in this paper of scaled gates should be mentioned (these are similar to the analog blocks in this paradigm).

Report

The topic of digital quantum simulation is enhanced via the analysis of strategies for scaling up the simulated models via commuting terms which can be implemented in parallel, perhaps by means of native interactions which are named scaled gates here. Moreover, a novelty I find here is the time of the simulator analysis, which is interesting, especially in terms of the mentioned "scaled gates" which can reduce the implementation time of the simulator. This is also related to the analog blocks in digital-analog quantum simulations referred to in Ref. (8) in the submitted version, and perhaps this could be more explicitly mentioned, even though I think that the time analysis is rather new in this context, and possibly useful.

Several typos throughout the paper should be corrected. A clearer mention of digital-analog quantum simulations as connected to the current concept in this paper of scaled gates should be mentioned (these are similar to the analog blocks in this paradigm).

Recommendation

Ask for minor revision

---

## Round 2 · Author Response

I'd like to thank the anonymous referee for their reading of the paper and for the helpful comments.

  • I have strengthened the link between scaled gates and the concept of digital-analog quantum simulation of Ref. [8] at several places in the text.

  • Also, I have corrected an embarrassingly large number of typos (thanks for the hint)

I hope that with these changes, the paper can be accepted for SciPost Core.

---

## Round 2 · List of Changes

• Added the text: "The concept of scaled gates is also related to the analog blocks in digital-analog quantum simulations [8]." towards the end of the third paragraph of the Introduction.

  • Added the text: "(see also the concept of digital-analog simulation [8])" in the discussion of scaled gates in Section III, bottom of page 3.

  • Added the text: ", similar to the analog blocks in digital-analog simulation [8]." in the Section V, 3rd paragraph (bottom of page 4).

  • Corrected several typos throughout the manuscript.

The reference occurring in these changes is:

[8] L. Lamata, A. Parra-Rodriguez, M. Sanz, and E. Solano, Digital-analog quantum simulations with superconducting circuits, Advances in Physics: X 3, 1457981 (2018), https://doi.org/10.1080/23746149.2018.1457981

---

## Editorial Decision

published